# The Nature and Extent of Plasmid Variation in *Chlamydia trachomatis*

**DOI:** 10.3390/microorganisms8030373

**Published:** 2020-03-06

**Authors:** Charlotte A. Jones, James Hadfield, Nicholas R. Thomson, David W. Cleary, Peter Marsh, Ian N. Clarke, Colette E. O’Neill

**Affiliations:** 1Clinical and Experimental Sciences, Faculty of Medicine, University of Southampton, Southampton General Hospital, Southampton SO166YD, UK; charlotte_a_jones@hotmail.com (C.A.J.); d.w.cleary@soton.ac.uk (D.W.C.); inc@soton.ac.uk (I.N.C.); 2Vaccine and Infectious Disease Division, Fred Hutchinson Cancer Research Center, Seattle, WA 98109-1024, USA; jhadfiel@fredhutch.org; 3Pathogen Genomics, The Wellcome Trust Sanger Institute, Wellcome Trust Genome Campus, Hinxton, Cambridge CB10 1SA, UK; nrt@sanger.ac.uk; 4Public Health England, Porton Down, Wiltshire SP40JG, UK; peter.marsh@phe.gov.uk

**Keywords:** *Chlamydia trachomatis*, plasmid, sequencing, genetic variation, evolution, diagnostics

## Abstract

*Chlamydia trachomatis* is an obligate intracellular pathogen of humans, causing both the sexually transmitted infection, chlamydia, and the most common cause of infectious blindness, trachoma. The majority of sequenced *C. trachomatis* clinical isolates carry a 7.5-Kb plasmid, and it is becoming increasingly evident that this is a key determinant of pathogenicity. The discovery of the Swedish New Variant and the more recent Finnish variant highlight the importance of understanding the natural extent of variation in the plasmid. In this study we analysed 524 plasmid sequences from publicly available whole-genome sequence data. Single nucleotide polymorphisms (SNP) in each of the eight coding sequences (CDS) were identified and analysed. There were 224 base positions out of a total 7550 bp that carried a SNP, which equates to a SNP rate of 2.97%, nearly three times what was previously calculated. After normalising for CDS size, CDS8 had the highest SNP rate at 3.97% (i.e., number of SNPs per total number of nucleotides), whilst CDS6 had the lowest at 1.94%. CDS5 had the highest total number of SNPs across the 524 sequences analysed (2267 SNPs), whereas CDS6 had the least SNPs with only 85 SNPs. Calculation of the genetic distances identified CDS6 as the least variable gene at the nucleotide level (d = 0.001), and CDS5 as the most variable (d = 0.007); however, at the amino acid level CDS2 was the least variable (d = 0.001), whilst CDS5 remained the most variable (d = 0.013). This study describes the largest in-depth analysis of the *C. trachomatis* plasmid to date, through the analysis of plasmid sequence data mined from whole genome sequences spanning 50 years and from a worldwide distribution, providing insights into the nature and extent of existing variation within the plasmid as well as guidance for the design of future diagnostic assays. This is crucial at a time when single-target diagnostic assays are failing to detect natural mutants, putting those infected at risk of a serious long-term and life-changing illness.

## 1. Introduction

*Chlamydia trachomatis* is a long-established cause of infection in humans, with mentions of the ocular form of the disease dating back to ancient Egyptian times [1]. Whilst trachoma has been eliminated from most Westernised countries, *C. trachomatis* continues to be the leading cause of preventable blindness in developing countries and remains the most common bacterial sexually transmitted infection worldwide. Through its obligate intracellular developmental cycle, *C. trachomatis* receives a continuous supply of nutrients and energy from the host cell it inhabits, but the caveat to this is the constant threat of the host immune response. Therefore, *C. trachomatis* has evolved an arsenal of evasion tactics preventing its detection and consequent elimination (as recently reviewed by Wong et al., 2019 [2]). The ready availability of nutrients has resulted in the gradual mutation and loss of biosynthetic genes that are surplus to requirements in the intracellular environment, and as a result of this adaptation process *C. trachomatis* has a very small genome at 1.04 Mb [3]. Despite this, most members of the Chlamydiacae also contain a 7.5 Kb plasmid [4], which is maintained at a low copy number of around four to seven times that of the chromosome [5]. The plasmid consists of eight coding sequences (CDS) (the functions of each plasmid CDS are summarized in Table 1) and two small antisense RNA sequences (sRNA) [6]. sRNA-2 is antisense to CDS2 and the most abundantly expressed feature of the plasmid, with up to 100-times the expression level of any plasmid-encoded CDS; sRNA-7 is antisense to CDS7 and whilst having relatively high expression levels when compared to the CDS features [7], this was later found to be 12-fold lower than that seen for sRNA-2 [8]. The chlamydial plasmid also has a 22 bp repeat region that serves as the plasmid’s origin of replication [9,10,11,12], similar to that seen in the iteron plasmids of many other gram negative bacterial species (for a full explanation of iteron plasmid function see Konieczny et al., 2014 [13]). Iteron sequences play a crucial role during replication initiation as they are recognized and bound by the (usually plasmid-encoded) Rep protein, which is the first step in the plasmid replication process. Variations within the iteron sequence can abolish this recognition or binding, so sequence conservation of the iteron is important. Iterons are also critical for control of the plasmid copy number [14] due to the “hand-cuffing” mechanism in which separate plasmid molecules are coupled via the ATP-dependent DNA helicase Rep protein, resulting in steric hindrance and prevention of subsequent rounds of replication [15]. In the chlamydial plasmid, there are usually four repeat sequences although in the plasmid of *C. trachomatis* strain SW5, only three repeats were identified with no apparent effect on plasmid copy number [16]. This suggests that three copies are sufficient for plasmid replication, or that a different mechanism of copy number control exists in this species.

The *C. trachomatis* plasmid is well conserved [17,35], with only 83 variable sites identified among the plasmids of the 11 strains analysed [16]. This conservation, and the near-ubiquity of the plasmid among clinical isolates suggests the plasmid has a role in infectivity. Plasmids are usually accessory to bacterial survival but confer a selective advantage, such as through the carriage of antibiotic resistance or virulence-associated determinants [36]. The chlamydial plasmid contains no homology to known antibiotic resistance genes, and the plasmid is dispensable in vitro [37]. However, the *C. trachomatis* plasmid is a virulence determinant in the natural host [38,39,40,41,42] and as a result, naturally occurring plasmid-free variants are increasingly rare [37,43,44]. Plasmid-encoded gene CDS5, which encodes the secreted Pgp3 protein [23], was identified as the key plasmid component for in vivo fitness and the induction of inflammatory responses [27]. CDS6 has a key role in regulating Pgp3 expression and also glycogen accumulation in the inclusion, but, like CDS5, is not involved in plasmid replication or maintenance. CDS5 and CDS6 can therefore be experimentally deleted with no apparent detrimental effect to fitness in cell culture [45,46]. Such knowledge is useful when designing plasmid-based shuttle vectors for transformation of chlamydia. Chlamydial shuttle vectors generally consist of the entire chlamydial plasmid and some *E. coli*-derived components, including an origin of replication (to facilitate genetic manipulation in *E. coli*) and antibiotic resistance marker(s). However, as a result these shuttle vectors tend to exceed 10 Kb in length, which may contribute to the extremely low transformation efficiency. A comprehensive understanding of the processes of plasmid maintenance and replication is critical in the design of new (minimal) vectors that could improve transformation efficiency. As such, vectors lacking CDS5 and CDS6 have already been engineered [45,46], and it is possible that other nonessential genes may be removed to help to further streamline the vector. An understanding of which genes are essential has proven invaluable in construction of the first chlamydial suicide vector, in which CDS8 (essential for plasmid maintenance or replication) was placed under control of an inducible promoter [47].

Understanding the function, diversity and stability of the *C. trachomatis* genome is also essential when choosing appropriate targets for molecular diagnostics. Until the advent of DNA amplification methods, detection of *C. trachomatis* infection relied upon diagnosis by tissue culture. However, in the last decade nucleic acid amplification technologies (NAAT) have taken over as the gold-standard approach to chlamydial diagnostics. Diagnostic target sites are chosen to avoid regions of known variation, and in doing so maximise chances of detecting all existing and future variants. Therefore, target sites should ideally be in a region that is essential for the organism to function, to minimise the chance of future evolutionary changes to the sequence. This was exemplified by the emergence of a urogenital mutant strain in Sweden that contained a 377 bp deletion in plasmid gene CDS1 [48]. This gene contained the single diagnostic target site of the prevailing diagnostic assays at the time; the choice of this target sequence appears to have been somewhat arbitrary, in a gene that shows a relatively high degree of variation. The mutation of CDS1 resulted in clonal expansion of that mutant due to missed diagnoses. This, combined with the frequent absence of clinical symptoms, has serious health implications due to the long-term complications associated with untreated chlamydia infections—such as pelvic inflammatory disease, ectopic pregnancy and infertility. Thus, choice of diagnostic target site is of paramount importance, and a complete understanding of plasmid diversity holds the key to designing a successful diagnostic assay.

In the present study we investigated plasmid diversity among 524 plasmid sequences sampled from 21 countries over a 50-year timescale (1957–2012). Previous analyses on these sequences focused solely on large-scale recombination events and plasmid swapping between lineages, which revealed occasional evidence of horizontal transfer of plasmids between clades [49]. The present study is a more detailed analysis of individual coding sequences within these plasmids with the aim of building on previous knowledge on the most and least conserved sequences of the plasmid, as well as to assess the nature of the mutations identified. This is the most comprehensive study of *C. trachomatis* plasmid diversity to date, as the sequences analysed are from a worldwide distribution and include both modern and historic samples.

## 2. Materials and Methods

### 2.1. Samples

The plasmid sequences used in this study were generated from whole genome sequencing data described previously: a total of 563 samples were collected from 21 countries, sampled between 1957 and 2012 [49]. Of these, 524 were analysed in this study—the remaining sequences were discounted due to low coverage or missing plasmid data. Sample types included clinical swabs, cell culture and yolk sacs from embryonated hens’ eggs (see Appendix A for sample metadata). The dataset encompassed all currently described genotypes of *C. trachomatis*: A (47 sequences), B (14), Ba (1), C (7), D (56), E (143), F (60), G (55), H (18), I (12), Ia (2), J (18), K (36), L1 (20), L2 (9), L2b (25) and L3 (1). 

### 2.2. DNA Extraction and Sequencing

DNA was extracted using immunomagnetic separation (IMS) using the MAb29 monoclonal antibody against chlamydial LPS (Chlamydia Biobank, Southampton, UK) [50], both with and without multiple displacement amplification (MDA) [51]; the SureSelect enrichment system (Agilent, Santa Clara, CA, USA) was used for direct sequencing from clinical swabs, as previously described [52].

Once extracted, DNA was sheared to an average length of 300 bp. Sequencing was performed using multiplexed Illumina HiSeq (Illumina, San Diego, CA) at the Wellcome Trust Sagner Institute (Cambridge, UK), using paired read lengths of 75–100 bp. Reads were mapped to an appropriate reference plasmid (ocular: A/HAR-13; lymphogranuloma venereum (LGV): L2/434; urogenital T1 clade: F/SW4; urogenital T2 clade: D/UW-3), and checked for quality and coverage as previously described [49].

### 2.3. SNP Detection

Single nucleotide polymorphism (SNP) detection was performed as previously described [35] with Wellcome Trust Sanger Institute (Cambridge, UK) pipelines, and an alignment of all sequences was generated using stringent parameters by mapping these against an artificial plasmid sequence generated from the Swedish New Variant (the most mutated *C. trachomatis* plasmid identified to date), which includes the 44 bp duplication, but the 322 bp deletion is restored; this approach elicits the maximum variation from the plasmid dataset. Each SNP in the alignment was documented by hand and for randomly selected SNPs, reads were checked as an additional quality control measure. Deletions identified in the alignment were validated against sequencing coverage plots. Summary tables of SNPs for each CDS were produced; see Appendix A. 

### 2.4. dN/dS Ratio

Estimations of selection for each codon were calculated using the dN/dS ratio. For each codon, estimates of the numbers of inferred synonymous (s) and nonsynonymous (n) substitutions are presented along with the numbers of sites that are estimated to be synonymous (S) and nonsynonymous (N). These estimates are produced using the joint Maximum Likelihood (ML) reconstructions of ancestral states under a Muse–Gaut model [53] of codon substitution and Felsenstein 1981 model [54] of nucleotide substitution. For estimating ML values, a tree topology was automatically computed. The test statistic dN– dS is used for detecting codons that have undergone positive selection, where dS is the number of synonymous substitutions per site (s/S) and dN is the number of nonsynonymous substitutions per site (n/N). A positive value for the test statistic indicates an overabundance of nonsynonymous substitutions. In this case, the probability of rejecting the null hypothesis of neutral evolution (*p*-value) is calculated [55,56]. Values of *p* less than 0.05 are considered significant at a 5% level. Normalized dN–dS for the test statistic is obtained using the total number of substitutions in the tree (measured in expected substitutions per site). It is useful for making comparisons across data sets. Maximum likelihood computations of dN and dS were conducted using HyPhy software package [56]. Codon positions included were 1st + 2nd + 3rd + Noncoding. All positions with less than 95% site coverage were eliminated. That is, fewer than 5% alignment gaps, missing data, and ambiguous bases were allowed at any position. Evolutionary analyses were conducted using the Molecular Evolutionary Genetic Analysis software, MEGA7 [57] (version 7.0.26, Pennsylvania State University, University Park, PA.). Analyses for each gene are presented in Appendix A.

### 2.5. Maximum Distances between Plasmid Sequences

Nucleotide alignments for each CDS were made in MEGA7 and estimates of average evolutionary divergence over all sequence pairs were computed. The number of base differences per site from averaging over all sequence pairs are shown. Standard error (SE) estimate(s) are shown in Table 3 and were obtained by a bootstrap procedure (1000 replicates). The analysis involved 524 nucleotide sequences. Codon positions included were 1st + 2nd + 3rd + Noncoding. All ambiguous positions were removed for each sequence pair. Evolutionary analyses were conducted in MEGA7 [57].

### 2.6. Variations in Number of 22 bp Repeats

22 bp repeat sequences are thought to act as the origin of replication for the plasmid [10,58]. Different numbers of repeats may impact on copy number of the plasmid; thus, to determine number of repeats for each sequence, the alignment of all plasmid sequences was visually inspected and repeats were counted. Repeats were classified as perfect (TTTGCAACTCTTGGTGGTAGAC) or imperfect (any variation of a perfect repeat). 

### 2.7. Clustering and Rarefaction Analysis of Plasmid Sequences

Aligned sequences were clustered using cd-hit-est v. 4.8.1 [59], using parameters -c 1.0 (100% identity threshold), -g (stringent clustering) and with all N bases masked. Clusters were then assigned to the relative plasmid lineages such that a matrix was produced of lineages x by clusters y. The R package vegan v. 2.5-6 [60] was then used to generate rarefaction curves for each plasmid lineage, and separately for the all plasmid clusters observed. Simpsons index (1-D) was calculated to illustrate cluster diversity within each plasmid lineage. 

### 2.8. Phylogenetic Reconstruction

Maximum likelihood analysis was performed on the dataset using TreeTime [61] using the Nextstrain platform, which consists of data curation, analysis and visualization components [49]. 

## 3. Results

### 3.1. Overall Plasmid Diversity

All SNPs identified in the plasmid dataset were recorded and are shown in Appendix A. A total of 199 SNP loci were found within genes which comprise 7005 of the total plasmid nucleotides, giving an intragenic SNP rate (SNPs per nucleotide in all coding sequences) of 2.84% (Table 2). There were an additional 24 SNP loci outside of coding regions, giving an overall plasmid SNP rate of 2.97%. The gene with the most variable sites was CDS3 (32 SNP loci) and the least was CDS6 (6 SNP loci); but as gene length varies considerably, a more informative figure is the SNP rate (number of SNP loci per nucleotide in that gene). Thus, the gene with the highest SNP rate was CDS8 (3.9%) and the lowest was CDS6 (1.94%). 

### 3.2. Evolutionary Distances

For the nucleotide sequence alignments, the overall mean distances for each CDS suggest that CDS6 is the least variable gene at the nucleotide level (*p* distance d = 0.001), and most variable gene is CDS5 (d = 0.007) (Table 3); the other genes were similar to the mean distance for the entire plasmid (d = 0.004). Analysis of the amino acid alignments agreed that CDS5 was the most variable gene, with a genetic distance ten-fold higher (d = 0.013) than that seen for the least variable gene at the amino acid level, which was CDS2 (d = 0.001); CDS6 was the second least variable gene in this analysis (d = 0.002).

### 3.3. dN/dS Ratio

The dN/dS test statistic was applied to each CDS independently to determine whether any codons are evolving due to selection (Appendix A). The normalised dN-dS test statistic was not significant for any codon within any of the CDS (*p* > 0.05 in all cases), suggesting that none of the genes significantly departed from neutrality so are not evolving due to positive or negative selective pressures.

### 3.4. SNP Characteristics

#### 3.4.1. Position of SNP within the Codon

The nature of intragenic SNPs determines the effect on the encoded protein. In this dataset, 62.9% of all intragenic SNPs occurred at the third base position, and in nearly all genes this was the most commonly mutated position (Table 4). However, CDS7 was an exception, as 58% of all SNPs were found to be in the second base position, 29% in the first position and only 13% in the third. Conversely, when considering only the position of SNP loci, irrespective of their frequency across the dataset, CDS7 had a normal distribution of SNPs throughout the codon, whereas in CDS1 53% of SNP loci occurring at the first base position, compared to 13% in the second position and 33% in the third position.

#### 3.4.2. Ratio of Synonymous to Nonsynonymous SNPs

The consequence of a SNP at a particular codon position and the nature of that SNP determines whether there will be an amino acid substitution or not. A SNP that results in an amino acid substitution is termed nonsynonymous, and “silent” SNPs that do not cause a change of amino acid sequence are termed synonymous. The percentage of SNPs that are nonsynonymous is informative when considering the importance of maintaining the coding sequence and indicates whether a gene is subject to evolutionary forces or not. Thus, the percentage of SNPs that were nonsynonymous was calculated for each gene (Table 2). The CDS with the lowest percentage of nonsynonymous SNPs was CDS2 (21%), and the highest was CDS7 (86%). Overall, 42% of intragenic SNPs were nonsynonymous.

#### 3.4.3. Amino Acid Substitution Characteristics

Furthermore, amino acids can be grouped by their characteristics, including whether they are polar, nonpolar, acidic or basic. For each of the nonsynonymous SNPs, the percentage which involved a change to an amino acid with different characteristics was calculated (Table 2). Of the nonsynonymous SNPs, 61.6% resulted in a change of amino acid characteristics, and as a percentage of all intragenic SNPs this figure is 25.9%. These SNPs have an increased likelihood of effecting protein function. CDS3 had the lowest percentage of nonsynonymous SNPs that resulted in an amino acid change of characteristics at 31.5%, whereas CDS6 had the highest percentage at 96.4%— however, it should be noted that CDS6 is the smallest gene at only 306 nucleotides long, with only 6 SNP loci in the gene, carried by 85 sequences; of these, 55 were nonsynonymous mutations.

#### 3.4.4. Occurrence of tri-allelic SNPs

The vast majority of SNP loci existed as two variants; however, there were 8 SNP loci for which three variants were possible (Appendix A). These tri-allelic sites existed in CDS 1, 3, 4, 5, 7 and 8, and none were found in CDS2 or CDS6. Most of the SNPs had a dominant variant, with the other variant existing in a single strain and therefore most likely represent sequencing errors. Two variants of tri-allelic SNPs were carried in multiple strains (at bases 5372 and 6065). CDS4 contained two tri-allelic sites; additionally, one of these sites formed the third base position (A, T or G) of a codon in which the first base was also variable (T or A), resulting in three possible amino acids (leucine, isoleucine or methionine). Furthermore, this codon was deleted in 71 of the sequences and so represents an unusual degree of polymorphism. To investigate this, each of the variants was mapped to an appropriate reference sequence and this hypervariable region was studied in detail. This revealed an area of “low complexity” in CDS4 which contains highly similar, repeated sequence motifs which have most likely confounded the automated assembly process—in all cases, the read coverage dropped suddenly at this locus, which supports this assertion. Therefore, to ascertain the true sequence at this locus PCR amplification and sequencing of each sample would need to be performed.

### 3.5. SNP Frequency at Specific Loci

SNPs at particular loci occurred at a range of frequencies in the dataset. Two of the variable loci contained a SNP in 287 of the sequences (these were present in CDS3 and CDS7), whereas 60 SNP loci only occurred in a single sequence (Figure 1). As stringent parameters were used in the SNP calling, these single occurrences most likely represent true events and not sequencing errors. Grouping of this data by CDS revealed that CDS5 had the highest frequency of SNPs, with 2276 SNPs across all sequences among the 28 SNP loci present in this gene (Table 2). Conversely, CDS6 had only 6 SNP loci and only 85 strains carried these SNPs. SNPs were distributed evenly across the plasmid, with no clustering observed (Figure 1). 

### 3.6. Identification of Premature and Delayed Stop Codons

The introduction of a premature stop codon, or conversely the loss of the natural end to a gene through mutation could have profound effects on the activity of the translated protein. In this dataset there were only three such occurrences, with two delayed and one premature stop codon (Table 5). In these cases, the impact on the length of the coding sequence was small, as the change in size of the CDS did not exceed a maximum of 4 codons deviation from the normal length. The delayed stop codons found in CDS1 and CDS4 were rare, occurring only 7 or 2 times in the dataset, respectively. The premature stop codon found in CDS4; however, occurred in 33 (6%) of the sequences and resulted in shortening of the protein by four amino acids.

### 3.7. Phylogenetic Reconstruction

The plasmid phylogenetic tree can be viewed at: https://nextstrain.org/community/jameshadfield/scratch/ct. The overall plasmid phylogeny is already described in Hadfield et al., (2017) [49], and the link to this interactive representation of the data is included here as it was not previously available and will help the reader visualise the relatedness of individual isolates described in the present study. 

### 3.8. Clustering and Rarefaction Analysis of Plasmid Sequences

A total of 182 clusters were identified from the 524 aligned sequences. The majority of clusters were represented by a single sequence (*n* = 132), with a further 43 clusters of 9 sequences. The largest cluster had 76 sequences (74 Lineage E plasmids with F_Fin106 and Ba_Apache2), followed by clusters of 38, 34 and 33. Figure 2 shows the rarefaction analysis of plasmids both as a total population and stratified by plasmid lineage. No asymptotes were observed in either case. Notable sampling bias is seen for Lineage E. With respect to the number of clusters identified, Lineages D and J were the most diverse, both having Simpsons 1-D of 0.93, with Lineages A and C being the least diverse, 1-D of 0.44.

### 3.9. Number of 22 bp Repeats

The number of 22 bp repeats may influence the plasmid copy number in *C. trachomatis* [17]. In this dataset, the majority of sequences (257, 49%) had four perfect 22 bp repeats; however, there were many sequences in which three perfect repeats were present, but the fourth was mutated to varying degrees (177, 34%). Some had lost more than half of the fourth repeat (Table 6). There were sporadic sequences with fewer repeats, and two sequences had none at all. However, upon visual inspection of read data, all sequences had at least 22 bp repeat sequences.

## 4. Discussion

### 4.1. Nature and Extent of C. trachomatis Plasmid Diversity

Advancements in genome sequencing technology over the past 20 years have shed light on many aspects of chlamydial biology and epidemiology, and with hundreds of whole genome sequences now in the public domain we have the ability to delve into the evolutionary history of *C. trachomatis* to a depth not previously possible. We now understand that modern lineages are the product of thousands of years of evolution rather than millions [49], and that the chlamydial plasmid has been vertically inherited throughout its evolutionary history, with very few instances of recombination or exchange between lineages [16,35,49,62]. Furthermore, through genetic manipulation and in vivo experiments, the role of each plasmid CDS in chlamydial virulence, regulation of gene expression and plasmid maintenance are gradually being revealed (Table 1). However, few studies have performed in-depth analyses on multiple *C. trachomatis* plasmid sequences [16,35,62,63], and most of these studies considered relatively few isolates (the largest study up until now included 157 sequences in their analysis [62]. In the present study, we analysed 524 *C. trachomatis* plasmid sequences, providing the largest in-depth study on chlamydia plasmid diversity to date. Analysis of this larger dataset has resulted in an increased capture of *C. trachomatis* diversity, with plasmid variation nearly three times greater (at 2.97%) than previously calculated [16,64]. The discovery of more variation seems an inevitable consequence of analysing increasingly large datasets due to the continual occurrence of point mutations. But, due to the relatively small availability of mutable sites in the plasmid (i.e., without incurring a fitness cost), we wondered if the large size of the current dataset and wide geographic and temporal distribution completes the picture of plasmid variation in *C. trachomatis*. In fact, it appears that this is not the case. The rarefaction curves generated upon resampling of data for the entire species and each genotype in isolation suggested that hitherto undescribed variation likely exists among *C. trachomatis* plasmids (Figure 2), and therefore further sampling may be warranted. The current availability of sequences is inherently biased towards the more common genotypes (such as E) (Figure 2), with some rarer types existing as sole surviving representatives (e.g., L3) or in very low numbers (e.g., C, I, H, L2). Therefore, the analysis of more examples of the rarer genotypes is likely to increase the diversity captured, highlighting the importance of collecting and preserving diverse isolates from divergent locations [22]. 

The majority of variation currently described within the plasmid consists of SNPs, with large-scale deletions and recombination events both being relatively rare [49] and indeed we did not identify any events such as these other than what has been previously described [16,35,49]. Additionally, CDS length was highly conserved with only one premature stop codon identified and two delayed stop codons, each causing only minor changes to the length of the CDS. This conservation of CDS length highlights the importance of the plasmid to chlamydial survival. 

Some diversity was identified within the replication origin of the plasmid, which comprises of (usually) four 22 bp repeat sequences [16]. Almost half of the sequences analysed here had four identical 22 bp repeats, and a further 34% of sequences had three complete and a fourth incomplete repeat sequence. This supports the notion that the 22 bp repeat sequence is important to plasmid maintenance and suggests that at least three repeats are required for efficient replication without affecting copy number [16]. However, in the present study a number of sequences had fewer repeats, with the third most common iteration being one perfect and one imperfect repeat (10%), particularly in E, F and L2b genotypes. However, the numbers are low, and it seems most likely that this is due to the high stringency of the assembly and alignment process rather than being a biological phenomenon; indeed, visual inspection of the read data showed that in the two isolates that apparently lacked a repeat region entirely, four repeat sequences were actually present. This observation was confirmed across five other randomly selected sequences that were reported as containing fewer than three repeats, and in each case, at least three repeat sequences were identified, highlighting the need to verify assembled genomes against read data.

Individual SNPs within coding sequences can have serious implications on the encoded protein due to alternations to the encoded amino acid sequence, and the position of the SNP within the codon is what determines its impact. The third base position of a codon is highly redundant, as around 67% of mutations at this locus are synonymous. The remaining 33% of mutations are nonsynonymous, but the physical characteristics of the encoded amino acid are maintained [65] and so the effect on the resulting protein will be minimal. The first codon position will result in a synonymous mutation in only 4% of cases, and second position mutations are always nonsynonymous; furthermore, most substitutions will always result in a change in amino acid characteristics at these sites [65]. As a result, one would expect the most common variable nucleotide position to be the third position, and indeed we found that across the entire plasmid 52.9% of intragenic SNP loci were at the third base position in the codon. However, CDS1 had an unusually high proportion of SNP loci in the first base position (51.6%), reflecting the previously identified redundancy of this gene in plasmid maintenance [16,17]. This is in stark contrast to its predicted functional homologue, CDS2, in which just 4% of SNP loci occurred at the first base position. Concordantly, CDS2 had the lowest percentage of nonsynonymous SNPs and can be considered the most functionally conserved gene of the plasmid, presumably due to its role in plasmid maintenance, confirming the much earlier study by Seth-Smith et al., (2009) [16]. Whilst variation in the CDS2 amino acid sequence was found to be uniformly low, variation in the nucleic acid sequence was found to be relatively high (fourth highest among the eight CDS). This may be explained by the existence of the overlapping sRNA-2 sequence, for which variations in sequence may form part of its function. Many sRNA molecules are involved in binding of mRNA, affecting their stability and/or translation [66]; others directly bind to protein transcription factors, affecting gene expression [67]; furthermore, the presence of SNPs in sRNA may alter the affinity to targeted mRNA and thus could have a significant effect on gene regulation [68]. Although the role of sRNA-2 is not yet understood, its expression levels at 12 h post infection [8] suggests an important role for sRNA-2 in regulation of genes important midway through the developmental cycle, such as RB replication, or possibly the early stages of RB-EB conversion [69,70,71].

In this study, each isolate was sampled at a single time point so there is no way of knowing whether that particular SNP variant later expanded to become a dominant clone, or if it vanished from existence due to deleterious effects—although predictions can be made based on their frequency in the dataset. The latter outcome possibly befell the most infrequently sampled SNPs, particularly the 60 SNPs that occurred only once among all isolates included in the study (Figure 1) which may represent transient events—although they may also indicate variation among isolates from under-sampled locations. The remaining SNPs were present in multiple sequences (Figure 1 and Appendix A). The very frequently sampled SNPs were branch-specific (no homoplasic SNPs were identified in this dataset), occurring early in the evolution of *C. trachomatis.* These became fixed either through chance (i.e., they are irrelevant to survival) or through offering a selective advantage to those strains carrying them. Tissue tropism-determining SNPs are well documented in the chlamydial chromosome (see reviews by [72,73,74]) but few have been identified in the plasmid [16]. Indeed, we did not identify any SNPs that consistently differentiated ocular from urogenital trachoma isolates; the SNP found to be unique to ocular strains by Seth-Smith et al. (2009) [16] was also found to occur in sporadic genotype G (G_S4658 and G_Ar246) and J (J_UK583676, J_UK35672, J_Soton72 and J_S42) isolates in the present dataset (these sequences were not available at the time of the earlier study). This reflects the close relatedness between these and trachoma (serovar A) plasmids previously noted [62], and may result from plasmid-swapping between strains, or recombination between plasmids; there is prior evidence of this from the present dataset [49]. Also, there is precedence for recombination between ocular and urogenital isolates [75] so the opportunity for genetic exchange must exist.

However, 30 SNPs were identified that consistently differentiated the LGV biovar from urogenital or ocular strains. Of these, 15 were synonymous mutations and so may have been retained through random genetic drift early in the evolutionary history of chlamydia, (i.e., at the point of divergence between trachoma and LGV lineages), and these mutations were then carried passively through subsequent evolutionary events. Possible exceptions are those synonymous mutations that fall within the two sRNA sequences overlapping CDS2 and CDS7, which may have an effect on secondary structure or mRNA binding. The 15 nonsynonymous mutations identified that divide LGV strains from the trachoma biovar are more likely to have arisen through natural selection due to the potential biological consequences of changes to amino acid sequences. Seven of these nonsynonymous SNPs occurred in CDS5 (Appendix A). The gene product of CDS5, Pgp3, is the only plasmid-encoded protein secreted into the inclusion lumen and cytosol [23]. Pgp3 may be important in host cell invasion [21], and has recently been identified as being an inhibitor of apoptosis in cell culture, via activation of the PI3K/AKT signalling pathway [32]. This interaction of Pgp3 with host cell signalling pathways suggests that CDS5 may be subject to immune selection, and the accumulation of LGV-specific nonsynonymous SNPs in this gene may suggest a role in LGV tropism, a notion supported by the five-fold higher expression of Pgp-3 in LGV compared to ocular strains [8]. Additionally, the crystal structures of Pgp3 from a urogenital (serovar D) and LGV (L1 440) strain have been resolved [21,22]. Pgp3 differs in structure between the biovars, with nine amino acid changes being identified between the two strains, resulting in the LGV version of Pgp3 occupying a different space group to that of the serovar D Pgp3 protein [22]. Across the present dataset, seven of the nine previously identified LGV-specific changes are maintained, but two amino acid replacements (T39K and D86N) also occur in many ocular and urogenital trachoma strains so are unlikely to contribute to LGV tropism. The two amino acids marked as being functionally significant in receptor binding (phenylalanine at amino acid site 6, and tryptophan at site 234) [22] are conserved across all sequences in the present dataset; however, a complete understanding of how the Pgp3 structure affects its biological function remains to be determined. 

Evidence that these CDS5 mutations have become fixed in the LGV biovar due to natural selection has yet to be provided, although attempts at investigating signs of selection in the *C. trachomatis* plasmid have been made using the dN/dS ratio [8,62]. Both studies found that although CDS5 had a dN/dS ratio indicative of positive selection (dN/dS > 1), this value was not statistically significant. As the present dataset is much larger than those previously analysed, we wondered whether the dN/dS ratio would tip towards statistical significance as a result of the additional sequences included in the analysis. But once again, none of the codons of any plasmid CDS, including CDS5, were found to significantly depart from neutrality. This is surprising given the near-ubiquitous carriage of the plasmid by clinical strains of *C. trachomatis* and the conservation of most of the plasmid-borne genes among diverse isolates; it seems highly unlikely that this has occurred by chance. However, it should be acknowledged that when the dN/dS ratio was developed, it was not intended for within-species comparisons; rather, the test was developed to analyse representative sequences of divergent species, where mutations are considered to be fixed [53]. When using this analysis for within-species data (where many mutations are transient in nature), this underlying assumption is violated [76]. As a result, the test is too conservative and results may be misleading [76,77]. Further work is needed before firm conclusions can be drawn about the effect of selective pressure on *Chlamydia* plasmid evolution. 

### 4.2. Implications on Diagnostic Target Choice

Modern diagnostic approaches are mainly based on nucleic acid amplification techniques, as these are highly sensitive and specific to the targeted organism. However, a complete understanding of the variation at the chosen target sites in the genome is essential for the continued efficacy of the test. In addition to the case of the Swedish New Variant, where a 344 bp deletion in the plasmid resulted in elimination of the single target site of a major diagnostic assay [78], there is now a second example where mutation of a single diagnostic target site has led to large-scale false-negative reporting of *C. trachomatis* [79]. The Finnish New Variant escaped detection by the Aptima Combo-2 test, which targets the 23S rRNA, but remained detectable by the Aptima CT test, which targets a sequence within the 16S rRNA sequence. The global distribution of the Finnish New Variant has yet to be determined, but it has also been detected in Sweden and further reports seem likely [80]. It is not known whether these isolates were imported from Finland or represent a separate evolutionary event—the former seems more likely given the close proximity of the two countries, but the latter cannot be ruled out until further investigations are completed. These examples of diagnostic failure highlight the importance of building a thorough understanding of target stability prior to employing a particular diagnostic target, with a focus on future stability being of paramount importance. Plasmid DNA tends to be present in multiple copies, thus targeting sequences within the plasmid affords a greater sensitivity of detection than chromosomal sites. However, the plasmid is accessory to survival; plasmid-free *C. trachomatis* isolates have been detected in clinical samples, although they are extremely rare [37,43,44,81,82]. It has previously been suggested that the plasmid might be a poor diagnostic target due to the opportunity for homologous recombination between plasmids and exchange of plasmids between isolates [35]; however, these events are infrequent. Accordingly, diagnostic tests employing dual-target assays that target both chromosomal and plasmid sequences should be preferentially considered to mitigate the risk of target deletion or plasmid loss, whilst retaining high sensitivity for low-level infections [83,84,85].

The relative stabilities of potential plasmid-based diagnostic targets were assessed by analysing the degree of sequence diversity at those sites. The first study to analyse multiple plasmid sequences identified CDS2 as being the most conserved gene at the nucleotide level [16]. However, more recently, CDS6 was identified as being the most highly conserved plasmid gene [62], and results from the present study support the latter study. Firstly, the SNP rate was determined by simply comparing each CDS on the number of SNP loci, normalized by dividing this by the total number of nucleotides in the gene. We found that CDS8 had the highest SNP rate (3.9%) whereas CDS6 had the lowest (1.94%), with CDS2 falling roughly in the middle, when considering only the number of SNP loci within the gene. A SNP that occurs frequently in the dataset is not necessarily more informative on evolutionary processes than a SNP that occurs infrequently, as a frequent SNP may be present in an over-represented genotype or may have occurred early in the evolutionary history of *C. trachomatis* with no effect on fitness. Such a SNP may be passively carried through subsequent generations. Nonetheless, this type of site may be informative for diagnostic target selection, as if it does not impair fitness it may have more chance of reverting to the ancestral state, which may result in reduced affinity between target site and diagnostic probe. In fact, the calculation of genetic distance (which considers the number of sequences carrying a SNP at each locus) found that whilst CDS6 retained the lowest value (d = 0.001), whereas the most variable gene was CDS5 (d = 0.007). Taken together, this suggests CDS6 may be a good choice for selection of diagnostic target sites. However, CDS6 is nonessential for stable plasmid maintenance in tissue culture [46], and its necessity in vivo has not yet been assessed. Nonetheless, CDS6 encodes the pgp4 protein, which has a role in the plasmid’s ability to accumulate glycogen [46] and is the sole regulator of pgp3 [19,86] and other virulence associated genes, suggesting CDS6 is important in the context of the natural host. Along with the presence of only 6 variable sites throughout the gene and the relative rarity of these SNPs in the dataset, this suggests that selective pressure exerted by diagnostic detection will be overcome by natural selection, resulting in its continued low variability. A disadvantage of using this gene would be the relative shortness of the coding sequence at only 309 bp in length, which may present challenges in optimizing primer design. A compromise between pragmatism and sequence stability needs to be reached when choosing optimal diagnostic target sites. 

A potential alternative diagnostic target to CDS6 may be offered by CDS3. Although CDS3 had the highest number of SNP locations, it had the second lowest SNP rate, once the length of the CDS has been taken into account (Table 2). CDS3 is the longest of the plasmid CDS features, which provides more options for the design of optimal primer pairs. Furthermore, the function of CDS3 has been assigned for some time and it is known to be essential for stable plasmid maintenance [19]. Thus, CDS3 may provide a useful alternative for assay design, if conserved sequences within CDS3 are targeted. This finding reflects the change implemented by Abbott Laboratories, who introduced the Abbott RealTime CT/NG assay, a dual plasmid assay combining primers for CDS1 and CDS3, replacing the single target Abbott m2000 assay which failed to detect the nvCT strain [87]. However, the continued use of CDS1 as a diagnostic target site is questionable given the high degree of variation seen in this gene in the present study and others [8,16,35,49,62].

The presence of 30 biovar-specific SNPs identified in this study may be useful for diagnostic purposes and could aid in identifying LGV infections in the clinic. This is important in the choice of treatment regimens, which differ due to a delayed antibiotic cure rate of LGV when compared to urogenital chlamydia infection [88,89]. Melt curve analyses using probes that target LGV-specific SNPs have been designed to discriminate LGV infections from urogenital strains, based on SNPs within the *ompA* and *pmpH* genes [90,91], but with the increased sensitivity afforded by the multi-copy plasmid, this information could offer a useful alternative.

## 5. Conclusions

This study describes the largest in-depth analysis of the *C. trachomatis* plasmid to date, mined from whole genome sequences spanning fifty years and from a worldwide distribution. Nevertheless, it seems that so-far undescribed plasmid diversity exists, highlighting the need for the continual collection and long-term preservation of representative isolates, which is best achieved through dedicated, accessible collections such as the Chlamydia Biobank (www.chlamydiabiobank.co.uk). The availability of genome sequenced live isolates promotes reproducibility in research, is an important resource to newcomers to the field and ensures the long-term survival of key strains that may otherwise be lost upon the closure of a lab or loss of a freezer.

A clear understanding of sequence stability is essential in the design of new targets for diagnostic assays and following the example of the Swedish New Variant it is curious that single-target assays are still being used for chlamydia detection. The Finnish Variant has now evolved as a direct result of diagnostic selection, enabling it to evade detection through mutation of the 23S rRNA sequence, a situation that could have been avoided by 1) choosing a more appropriate (less variable) target sequence and b) having a back-up target that is detected simultaneously to guard against mutation. A target within the plasmid is useful due to the presence of multiple copies, enhancing the sensitivity of the test. Within the dataset examined, we suggest suitable plasmid-based targets to aid in the design of new diagnostic assays, whilst recommending that these are applied in a dual-target approach with the other target lying within the chromosome to guard against target loss through plasmid elimination.

## Figures and Tables

**Figure 1 microorganisms-08-00373-f001:**
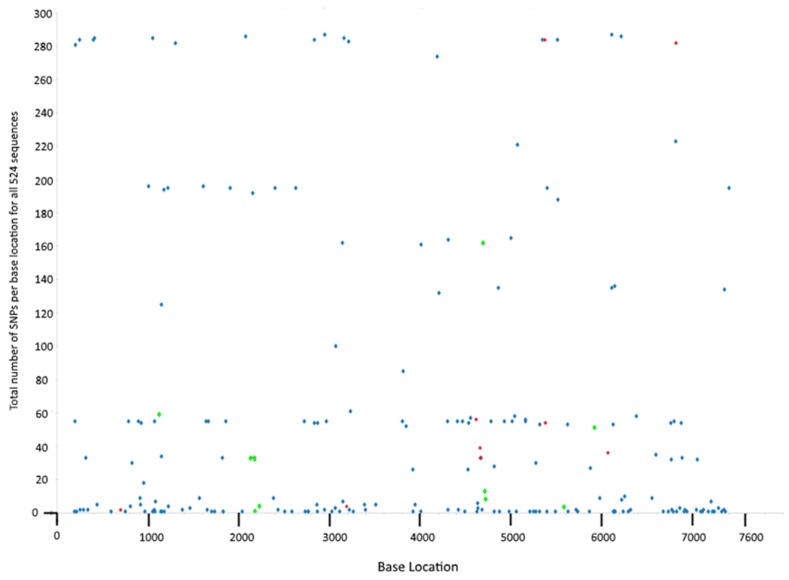
Number of sequences carrying each of the 199 SNP loci identified. Blue dots indicated bi-allelic SNP locations, red dots indicate tri-allelic SNP locations, and green dots are intergenic SNPs.

**Figure 2 microorganisms-08-00373-f002:**
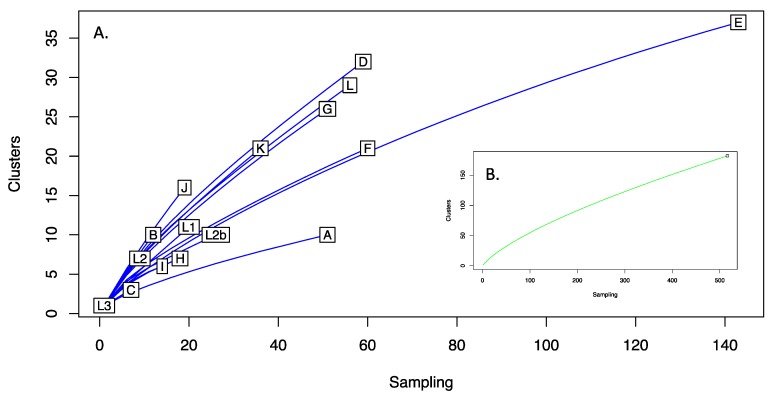
Rarefaction of *C. trachomatis* plasmid nucleotide sequences. Aligned sequences were clustered using cd-hit-est at 100% nucleotide identity. The diversity of each lineage, as measured by cluster discovery versus sampling effort (rarefaction), is shown for each plasmid lineage (**A**) and all plasmid sequences (inset (**B**)). No asymptotes were observed indicating sampling effort had not captured all the diversity within the *C. trachomatis* plasmid populations. Lineages A and C were the least diverse (L3 being represented by only one sequence), with D and J the most diverse.

**Table 1 microorganisms-08-00373-t001:** Functions of each plasmid CDS and RNA sequence.

CDS (Pgp)	Function of Encoded Protein	Summary of Current Knowledge and References
**1 (7)**	Plasmid Replication	Homologue of integrase, part of the family of phage proteins [17]. Role in regulation of plasmid replication [18]. Loss of function does not affect plasmid maintenance due to functional redundancy with CDS2 [16,17,19].
**2 (8)**	Plasmid Replication	Homologue of recombinase, part of the family of phage proteins; role in regulation of plasmid replication [17]. Critical for plasmid maintenance [19]. The main determinant of plasmid tropism [20].
**3 (1)**	Plasmid Replication	Homology observed with DnaB helicase proteins of *Escherichia coli* and *Salmonella typhimurium,* helicase involved in unwinding of double stranded DNA [11,17]. Critical for plasmid maintenance [19].
**4 (2)**	Function unknown	Required for plasmid maintenance [19].
**5 (3)**	Virulence protein	28 kDa protein [9]. CDS3-encoded Pgp (plasmid glycoprotein) 3 crystal structure resolved for genotypes D and L2 [21,22]. Dispensable for chlamydial growth in vitro [19]. Shown to be secreted from the chlamydial inclusion into the cytosol of the host cell [23,24]. Pgp3 from *C. muridarum* shown as major virulence factor responsible for hydrosalpynx induction in mice [25,26]. Strong immunogenic properties; purified protein stimulated macrophages to release inflammatory cytokines in the mouse model and acts as Toll-like receptor 4 (TLR4), suggesting a role in Chlamydia-induced inflammatory pathology [23]. Later shown to have an essential role in virulence and infectivity in vivo [27]. Pgp3 antibody was found to persist for at least 12 years post infection, suggesting a role for *C. trachomatis* Pgp3 serology in evaluating control programmes [28]. *C. trachomatis* Pgp3 neutralizes the antichlamydial activity of human cathelicidin LL-37 [29] and is essential for colonisation of the gastrointestinal tract [30] due to evasion of acidic barriers (in both stomach and vagina) [31]. Pgp3 expression is also shown to inhibit apoptosis via the PI3K-AKT-mediated MDM2-p53 axis [32].
**6 (4)**	Transcriptional regulation	Role in ability of *C. trachomatis* to accumulate glycogen [33]. Transcriptional regulation of the plasmid virulence protein Pgp3 and of chromosomal gene expression [19,25].
**7 (5)**	Regulation of partitioning and copy number	Partial homology to *E. coli* plasmid and phage encoded proteins, including SopA and ParA, which are involved in partitioning and copy number in *E. coli* [17]. Shown to negatively regulate some plasmid-dependant genes in *C. muridarum* [34]. Dispensable for chlamydial growth in cell culture [19].
**8 (6)**	Regulation of partitioning and copy number	Thought to function in conjunction with pCDS7 in a similar manner to that of the *sopA*/B and *parA*/B operons in *E. coli* [17]. The only CDS which has a homologue on the *C. trachomatis* chromosome, which is also present in plasmid free isolates [19]. Critical for plasmid maintenance [19].

**Table 2 microorganisms-08-00373-t002:** Number of SNP loci, rates, and characteristics of all SNPs in dataset. CDS, coding sequence. SNP rate is the number of intragenic SNP loci (base positions containing at least one SNP in any of the 524 sequences in the dataset), compared to the length of the CDS, expressed as a percentage. Total SNPs is the total number of SNPs across the dataset for all variable sites. Average number of SNPs per locus is the average number of SNPs per variable site across the CDS in question. Nonsynonymous SNPs per CDS are presented, along with the % of total SNPs. Nonsynonymous (N-S) SNPs involving a change of amino acid characteristics also shown as a percentage of the total number of nonsynonymous SNPs.

	Length (bp)	Number of Intragenic SNP Loci	SNP loci Rate (%)	Total SNPs	Average Number of SNPs per Locus	Non Synonymous (NS) SNPs (%)	NS SNPs Involving a Change of Amino Acid Characteristics (%)
**CDS**							
1	918	30	3.27	2012	67.06	762 (37.9)	436 (57.2)
2	993	26	2.62	1734	66.69	364 (21)	237 (65.1)
3	1356	32	2.36	2116	66.13	575 (27.2)	182 (31.7)
4	1065	27	2.54	1435	53.14	518 (36.1)	339 (65.4)
5	795	28	3.52	2267	80.96	1335 (58.9)	850 (63.4)
6	309	6	1.94	85	14.16	55 (64.7)	53 (96.4)
7	825	21	2.55	1072	51.05	928 (86.6)	722 (77.8)
8	744	29	3.90	1126	38.83	450 (40)	252 (56)
**Total**	7005	199	2.84	11,847	-	4987 (42.1)	3070 (61.6)

**Table 3 microorganisms-08-00373-t003:** The genetic distance d represents estimates of average evolutionary divergence over all sequence pairs for each coding sequence in isolation. The genetic distance of the entire plasmid is also shown for the nucleotide alignments, both coding and noncoding regions; amino acid comparisons cannot be made as not all nucleotides are in coding sequences. SE = standard error (1000 replicates). All numbers were rounded to 3 decimal places.

	Nucleotide Sequences		Amino Acid Sequences	
**CDS**	d	SE	d	SE
1	0.004	0.001	0.003	0.002
2	0.004	0.001	0.001	0.001
3	0.003	0.001	0.003	0.001
4	0.004	0.001	0.003	0.002
5	0.007	0.002	0.013	0.004
6	0.001	0.001	0.002	0.002
7	0.003	0.001	0.008	0.003
8	0.004	0.001	0.005	0.002
Plasmid	0.004	0.000	N/A	N/A

**Table 4 microorganisms-08-00373-t004:** SNP loci and total number of SNPs in relation to position within the codon. Percentage of bases at that position are shown in parentheses.

	Number of SNP Loci (%)				Total Number of SNPs (%)			
CDS	Base 1	Base 2	Base 3	Total	Base 1	Base 2	Base 3	Total
**1**	16 (53.3)	4 (13.3)	10 (33.3)	30	470 (23.4)	292 (14.5)	1250 (62.1)	2012
**2**	4 (15.4)	4 (15.4)	18 (69.2)	26	129 (7.4)	39 (2.2)	1566 (90.3)	1734
**3**	10 (31.25)	6 (18.75)	16 (50)	32	164 (7.8)	408 (19.3)	1544 (73)	2116
**4**	6 (22.2)	1 (3.7)	20 (70.1)	27	426 (29.7)	1 (0.1)	1008 (70.2)	1435
**5**	9 (32.1)	8 (28.6)	11 (39.3)	28	594 (26.2)	633 (27.9)	1040 (45.9)	2267
**6**	2 (33.3)	1 (16.7)	3 (50)	6	2 (2.4)	1 (1.2)	82 (96.5)	85
**7**	5 (23.8)	7 (33.3)	9 (42.8)	21	300 (28)	626 (58.4)	146 (13.6)	1072
**8**	4 (13.8)	8 (27.6)	17 (58.6)	29	114 (10.1)	196 (17.4)	816 (72.5)	1126
**TOTAL**	56 (28.1)	39 (19.6)	104 (52.3)	199	2199 (18.6)	2196 (18.5)	7452 (62.9)	11,847

**Table 5 microorganisms-08-00373-t005:** Details of premature and delayed stop codons collated from whole dataset.

Open Reading Frame	Base	Number of Sequences with SNP	Reference Code	SNP Change	Pre-AA	Post-AA	Type of Stop Codon (Premature or Delayed)	Change to Size of CDS
1	1080	7	TGA	GGA	STOP	G	Delayed	+3 codons
4	4667	33	GAA	TAA	E	STOP	Premature	−4 codons
4	4679	2	TAA	CAA	STOP	Q	Delayed	+1 codon

**Table 6 microorganisms-08-00373-t006:** Number of repeats in the 22 bp repeat region per genotype according to the automated alignment process. Imperfect repeats (any variation of a perfect repeat) are abbreviated to “imp.” in the table.

	Number of Repeats									
Geno-Type	Four Repeats	Three + imp.	Three Repeats	Two + imp.	Two Repeats	One + imp.	One Repeat	None + imp.	No Repeats	Total
**A**	0	45	0	0	0	2	0	0	0	47
**B**	13	1	0	0	0	0	0	0	0	14
**Ba**	0	1	0	0	0	0	0	0	0	1
**C**	5	2	0	0	0	0	0	0	0	7
**D**	32	11	5	1	1	6	0	0	0	56
**E**	66	58	0	1	0	16	0	0	2	143
**F**	27	20	3	1	0	9	0	0	0	60
**G**	33	10	5	2	1	4	0	0	0	55
**H**	16	2	0	0	0	0	0	0	0	18
**I**	7	3	0	0	1	1	0	0	0	12
**Ia**	0	0	0	0	0	2	0	0	0	2
**J**	8	7	1	0	0	2	0	0	0	18
**K**	19	8	2	2	2	3	0	0	0	36
**L1**	12	5	0	0	0	3	0	0	0	20
**L2**	7	2	0	0	0	0	0	0	0	9
**L2b**	12	2	0	1	0	9	0	1	0	25
**L3**	0	0	0	0	0	0	0	1	0	1
**Total**	257	177	16	8	5	57	0	2	2	524

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
