# Peer review of "The Nature and Extent of Plasmid Variation in Chlamydia trachomatis"

_microorganisms, 2020, doi:10.3390/microorganisms8030373_

Round 1

Reviewer 1 Report

This study investigated Chlamydia trachomatis plasmid polymorphism from 524 clinical samples and isolates obtained by the authors, as well as those already deposited in the GenBank. The use of such a large plasmid collection allowed in depth analysis of plasmid evolution and its worldwide distribution. The immediate application of this research can be recommendations on the plasmid-based diagnostics. The manuscript is clearly written, conclusions correspond to data obtained, and statistical treatment is appropriate. The major recommendation of the reviewer is to deposit all plasmid sequences to the GenBank database, and provide their accession numbers in the Supplementary data table 1.

The minor correction in the Supplementary data table 2 will be merging cells for the tri-allelic positions for the CDS1 (base 699) and CDS3 (base 3188), same as it is done for other CDSs. 

Author Response

Thank you for your helpful comments. It is a good point about making all of the sequence accession numbers available - I have now added a column to Supplementary Table 1 to include the ENA accession numbers (for the European Nucleotide Archive) as an alternative to Genbank. It would take months to submit everything to Genbank, whilst they are already freely available on the ENA database - I hope this is satisfactory. Also thank you for spotting the formatting error in supplementary table 2, I have now amended this.

Reviewer 2 Report

The manuscript is well written and coherent. All experiments conducted addressed the problems correctly. The study design and results are well presented. It is very informative about the subject covering all the plasmids present until 2012. Therefore I recommend the manuscript for publication in its present form.

1- A linear or circular map of the plasmid showing the structure of the CDS discussed and possibly highlighting the position of most informative SNPs and the location of the repeats ( even though the tables presented are clear enough).
2- One point that seems interesting is that the analysis was don on public data up until 2012. It is interesting why the authors did not include other sequencing data beyond that year ( even though the sample size analysed (almost 500) is enough for the conclusions stated.
3- the authors can also point out whether the plasmid contains an addiction system (TA) that was not reported in the manuscript and wether the genomes contain CRISPER systems ( should not have in this case) through the public database https://crispr.i2bc.paris-saclay.fr/

Author Response

Thank you for your helpful comments. We felt that a table was more informative than a diagram, although appreciate that a diagram does look nicer. We did not include data after 2012 as we had such a large number of sequences we felt it would capture the diversity of the plasmid - although it seems there is more diversity to be found, a more targeted approach to capture additional variation is needed, and could perhaps form a follow-up study. With regards to the possibility of a plasmid addiction system, my understanding was that this would mean that it is not possible for the bacterial cell to survive without a plasmid - chlamydia can survive both in vitro and in vivo without a plasmid, but it causes reduced pathogenicity in vivo. As this paper is about the plasmid and not the whole genome I have not included a discussion on CRISPR.